# Context-Seq: CRISPR-Cas9 targeted nanopore sequencing for transmission dynamics of antimicrobial resistance

Erica R. Fuhrmeister[1,2,3], Sooyeol Kim [ID][2], Shruteek A. Mairal [ID][4], Caroline McCormack [ID][2], Benard Chieng[5], Jenna M. Swarthout[6], Abigail Harvey Paulos[2], Sammy M. Njenga [ID][5] & Amy J. Pickering [ID][2,7,8] ✉

Precisely understanding how and to what extent antimicrobial resistance (AMR) is exchanged between animals and humans is needed to inform control strategies. Metagenomic sequencing has low detection for rare targets such as antibiotic resistance genes, while whole genome sequencing of isolates misses exchange between uncultured bacterial species. We introduce Context-Seq, CRISPR-Cas9 targeted sequencing of ARGs and their genomic context with long-reads. Using Context-Seq, we investigate genetically similar AMR elements containing the ARGs $bla_{CTX-M}$ and $bla_{TEM}$ between adults, children, poultry, and dogs in Nairobi, Kenya. We identify genetically distinct clusters containing $bla_{TEM}$ and $bla_{CTX-M}$ that are shared between animals and humans within and between households. We also uncover potentially pathogenic hosts of ARGs including *Escherichia coli, Klebsiella pneumoniae*, and *Haemophilus influenzae* in this study context. Context-Seq complements conventional methods to obtain an additional view of bacterial and mammalian hosts in the proliferation of AMR.

Antimicrobial resistance (AMR) is a global challenge that threatens to undermine modern medicine. It is estimated that in 2019, nearly 5 million deaths were associated with bacterial antimicrobial resistance[1]. The global burden of AMR disproportionately falls on low-income countries, where high rates of illness, unregulated antibiotic usage, and limited access to sanitation infrastructure contribute to the selection and spread of AMR bacteria[2,3]. Current approaches to control antibiotic resistance rely on antibiotic stewardship; however, this approach is difficult in low-income countries where unregulated antibiotic usage in humans and animals is common and the burden of infectious diseases is high. Between 2000 and 2015, antibiotic drug consumption rates increased by 77% in low- and middle-income countries, compared to a decrease of 4% in high-income countries[4].

AMR can be shared between humans, animals, and the environment. For this reason, AMR fits within the One Health framework, which integrates human, animal, and environmental health to tackle complex public health problems. AMR can spread through the dissemination of whole bacteria carrying resistance, as well as through horizontal gene transfer of mobile elements, including between benign and pathogenic bacteria. In addition, soil, water, and air are known environmental reservoirs of AMR[5-8]. From a molecular perspective, determining the genomic context of ARGs is critical for studying the proliferation of AMR as it can allow for the identification

[1]Department of Environmental and Occupational Health Sciences, School of Public Health, University of Washington, Seattle, WA, USA. [2]Department of Civil and Environmental Engineering, University of California, Berkeley, CA, USA. [3]Department of Civil and Environmental Engineering, University of Washington, Seattle, WA, USA. [4]Department of Chemical and Biomolecular Engineering, University of California, Berkeley, CA, USA. [5]Eastern and Southern Africa Centre of International Parasite Control, Kenya Medical Research Institute, Nairobi, Kenya. [6]Department of Civil and Environmental Engineering, Tufts University, Medford, MA, USA. [7]Blum Center for Developing Economies, University of California, Berkeley, CA, USA. [8]Chan Zuckerberg Biohub, San Francisco, CA, USA. ✉e-mail: pickering@berkeley.edu

of mobile elements, co-occurring genes, and host bacteria[9]. These additional pieces of information may yield insights on the mechanisms of exchange between reservoirs and the role of zoonotic transmission. In order to curb the spread of AMR, we need to be able to identify the most important transmission pathways (e.g., poultry, dogs, water, soil) in a given setting.

Current methods to investigate genetically similar AMR elements between reservoirs primarily rely on culturing and whole genome sequencing of isolates[10–12]. However, culturing only captures a small fraction of organisms, with a bias towards bacteria more fit for selective conditions. Culture-independent methods, such as metagenomic sequencing, require high sequencing depth to capture low-abundance targets such as ARGs[13]. In addition, untargeted metagenomic sequencing can waste millions of reads per sample with very low coverage of medically important ARGs. This abundance of data can be costly to store and require significant computing resources to analyze.

Targeted sequencing approaches are promising for studying AMR by enrichment of genomic regions of interest. Illumina probe capture is inherently limited by the read length (200–500 bps), which makes investigating ARGs in their genomic context infeasible. Recently, Cas9-based enrichment has been applied to short-[14] and long- read sequencing for clinical applications[15,16] targeting a variety of genes including *Klebsiella pneumoniae*-associated genes[17], ARGs[18], cancer-associated genes[15], and integrons[19]. In brief, extracted DNA is dephosphorylated followed by Cas9 cutting facilitated by guideRNAs. Sequencing adapters are selectively ligated to only the d(A) tailed ends that result from Cas9 cutting. For example, Cas9-guided adapter ligation was used to perform multiplexed detection of ARGs in human blood spots with Illumina short-read sequencing[14] and to investigate human alleles in breast tissue with Oxford Nanopore long reads[15]. The selectivity introduced through guideRNAs coupled with long-read sequencing that can capture long DNA fragments make this a promising approach to investigate ARGs within their genomic context.

In this study, we develop and optimize a Cas9 targeted sequencing assay to selectively sequence ARGs and their genomic context, hereby referred to as Context-Seq. We demonstrate the utility of Context-Seq by applying the method to detect the ARGs $bla_{CTX-M}$ and $bla_{TEM}$ in human (adult and child), poultry, and canine fecal samples collected from households in Nairobi, Kenya, to investigate genetic synteny in antimicrobial resistance elements.

## Results

### Samples and ARG target selection

In order to select relevant ARG targets for Context-Seq, we first analyzed available human and animal fecal samples collected from seven households in Nairobi, Kenya[20] using a Taqman Array Card to detect 14 ARGs and eight pathogen targets (Supplementary Table 1 and Supplementary Fig. 1). All samples were positive for *tetA*, *sul1*, and $bla_{TEM}$. $Bla_{NDM}$ and *mcr-1* were detected in canine and poultry samples only (*mcr-1*: 75% canine, 7% poultry; $bla_{NDM}$: 50% canine, 13% poultry). $Bla_{CTX-M\ Group\ 1}$, $bla_{CTX-M\ Group\ 9}$, $bla_{OXA-10}$, and $bla_{SHV}$ were detected in 46-100% of samples of each type. We selected two clinically relevant targets in high abundance ($bla_{TEM}$ and $bla_{CTX-M}$) and samples from four households that were positive for $bla_{TEM}$ and $bla_{CTX-M\ group\ 1}$ (Supplementary Table 2).

While many metagenomic approaches capture hundreds to thousands of resistance genes, not all resistance genes are clinically important[21]. Extended-spectrum beta-lactamases (ESBLs) genes are of high medical importance as they can confer resistance to most beta-lactams including cephalosporins[22]. $Bla_{CTX-M}$ is a globally distributed gene group where all alleles are considered ESBLs[23]. Common genotypes include $bla_{CTX-M-15}$ and $bla_{CTX-M-14}$[23]. $Bla_{CTX-M}$ alleles have been found in humans[24], animals[25], wastewater[26], and other environmental reservoirs[27]. They are also present in clinical isolates of *Escherichia coli*, *K. pneumoniae*, *Salmonella* species, *Pseudomonas aeruginosa*, and

other bacterial taxa[28]. Similarly, $bla_{TEM}$ is globally distributed, present in multiple reservoirs, and found in clinical isolates[25,29–32]. However, $bla_{TEM}$ alleles differ in phenotypic resistance conferred, ranging from penicillin resistance (e.g., $bla_{TEM-1}$) to ESBLs (e.g., $bla_{TEM-10}$)[33].

### Guide design tool

To design Cas9 guide RNAs, we utilized existing software (CHOPCHOP)[34] and developed a custom script to estimate off-target activity of guides in complex microbial communities, which is publicly available (https://github.com/Shruteek/Optimized-sgRNA-Design). Guides were selected based on high CHOPCHOP predicted efficiency, genomic location near the ends of the genes, and lower predicted off-target activity (Supplementary Table 3). Notably, predicted efficiency as well as off-target activity are based on empirical data[35] and may not be well representative of real systems. To facilitate capture of genomic context in both directions of a target ARG, different fractions of the sample DNA were cut by guides on the sense and antisense strands separately and then pooled. When comparing guides for $bla_{TEM}$, all pairs (sense and antisense) resulted in enrichment; there was, however, variation in enrichment based on the combined pair (Supplementary Fig. 2) and best performing guides were not necessarily predicted to have the highest on target activity and lowest off-target activity (Supplementary Table 3). Alleles targeted by guides are indicated in the supporting information (Supplementary Tables 4 and 5).

### Protocol optimization

We optimized long-read Cas9 enrichment, originally validated for variant detection in cell culture and human tissue samples[15], to detect ARGs in fecal and soil samples. We investigated these modifications on a test system comprised of DNA extracted from an *E. coli* isolate with $bla_{CTX-M-55}$ and $bla_{TEM-1}$ genes spiked into a composite sample of extracted DNA from Kenyan soil. Modifications evaluated included adaptive sequencing, multiple guides per target per strand, longer incubation time for Cas9 cleavage, and the addition of thermolabile Proteinase K. We also evaluated the impact of including two targets in the model system and in a human fecal sample.

Our final protocol enriched for two targets ($bla_{CTX-M}$ and $bla_{TEM}$) in two directions and added a thermolabile Proteinase K digestion to previously published methods (Fig. 1A). Adaptive sequencing, longer Cas9 digestion, and additional guides per target did not improve the assay performance (Fig. 1B) but utilizing thermolabile Proteinase K after the Cas9 digestion did (Fig. 1C). Inclusion of guides for both $bla_{CTX-M}$ and $bla_{TEM}$ decreased enrichment for both targets in our test system (Fig. 1B) and in the human fecal sample (Figs. 1D, E). There was a minor decrease in $bla_{TEM}$ when comparing coverage between one-target and two-target enrichment. The non-normalized coverage (mean ± std) was 1265 ± 253 for two targets and 1517 ± 337 for one target. Since $bla_{TEM}$ was the more abundant target, we processed samples enriching for both targets. All protocol modifications resulted in an enrichment of 7-15X coverage over untargeted methods (Fig. 1B).

Using our final protocol, we sequenced 13 fecal samples across four households (five human, three canine, and five poultry) on individual MinION flow cells (Supplementary Table 6). The percentage of reads that aligned to either $bla_{CTX-M}$ and/or $bla_{TEM}$ out of the total reads that passed quality filtering was 0.4% (range 0.02-2.01%). Of the reads that aligned to $bla_{TEM}$ across all samples, the average length was 4854 ± 1081 base pairs (bps) and of those that aligned to $bla_{CTX-M}$ the average length was 4381 ± 745 bps. As described in more detail in the methods, reads >1500 bps containing ARGs were clustered at 85% and polished if multiple reads were assigned to the same cluster. This generated one consensus sequence per cluster and resulted in an average of 36 ± 33 cluster for $bla_{TEM}$ and 3 ± 3 clusters for $bla_{CTX-M}$ per sample (Supplementary Table 7).

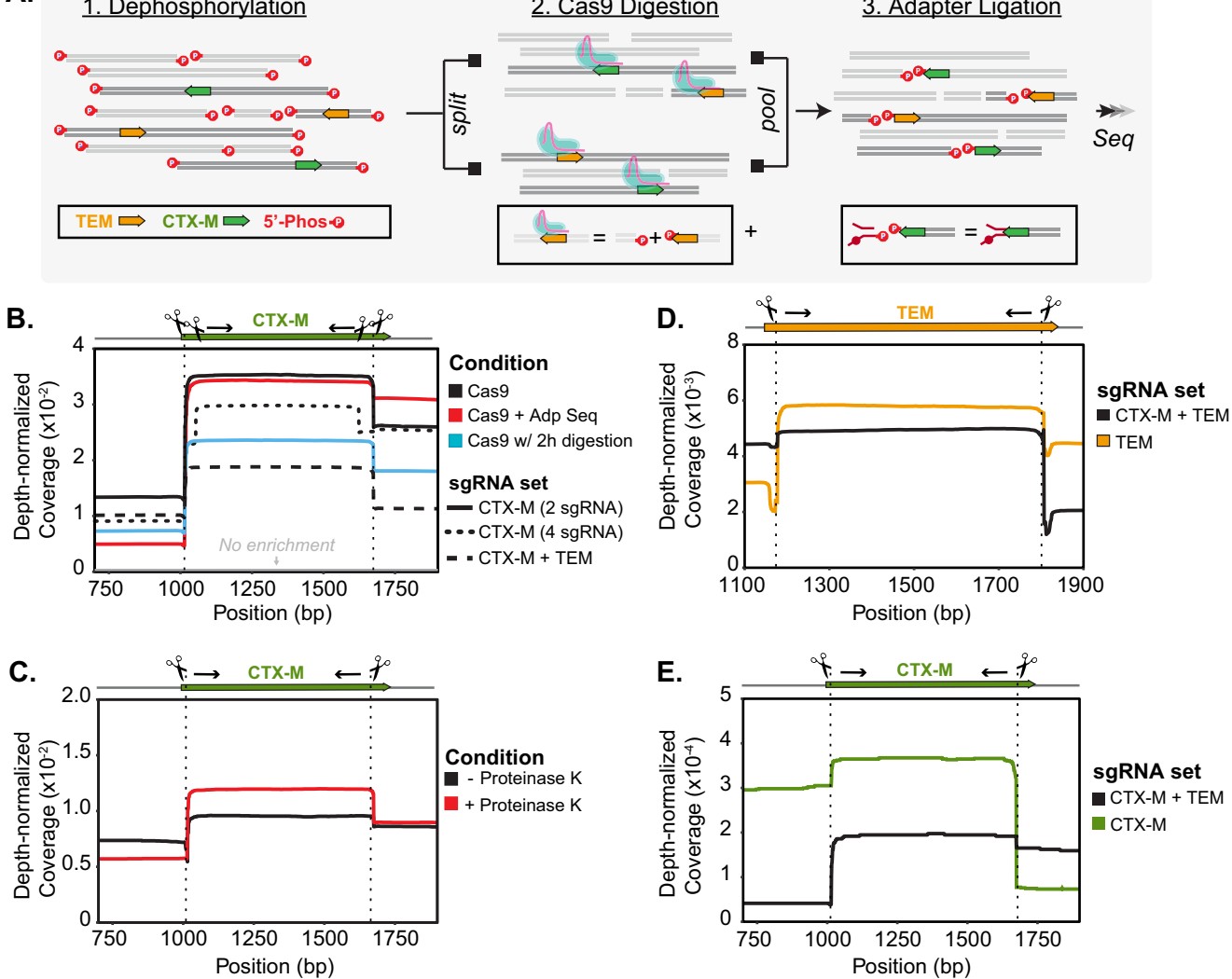

**Fig. 1 | Cas9 is used to selectively sequence DNA fragments containing $bla_{CTX-M}$ and $bla_{TEM}$. A** Schematic of Context-Seq workflow involving dephosphorylation, library splitting for Cas9 cutting on each strand, and adapter ligation. Phosphorylated ends are indicated with a red P. **B** Comparison between library preparation modifications (adaptive sequencing, longer Cas9 digestion, and additional guides) against no enrichment in a test system. Depth normalized coverage is calculated by dividing coverage by the total reads obtained per each sequencing run. **C** Comparison between conventional Cas9 enrichment protocol and the inclusion of Proteinase K following Cas9 digestion in a test system. **D** Normalized coverage of $bla_{TEM}$ in a human fecal sample comparing enrichment of $bla_{TEM}$ alone (yellow) and both $bla_{TEM}$ and $bla_{CTX-M}$ (black). **E** Normalized coverage of $bla_{CTX-M}$ in a human fecal sample comparing enrichment of $bla_{CTX-M}$ alone (green) and both $bla_{TEM}$ and $bla_{CTX-M}$ (black). Source data are provided in the Source Data file.

**Context-Seq enabled identification of ARGs and annotation of their surrounding genomic context.** Annotated sequences resulted in the expected cut pattern based on guide design. Sequences begin with an ARG cut on either the sense or antisense strand and contain additional annotated genes across variable long read lengths (Fig. 2). Sequences for $bla_{CTX-M}$ ranged from 1,662-17,369 bps and included mobile genes annotated as integration/excision (e.g., *tnpA, tnpR, hpaI, gin*), replication/recombination/repair (e.g., *dnaQ, repL, impB*), and phage (e.g., *ant, gp23, kilA, orf16*) (Fig. 2A). Sequences for $bla_{TEM}$ ranged from 1489–23,336 bps and included mobile genes annotated as integration/excision (e.g., *tnpA, tnpR, int*, IS6 family transposases), transfer (*mob, finO, traI*), replication/recombination/repair (e.g., *rop, repC, repM, parM*), and phage (*bof, cre, pacB*) (Fig. 2B). Co-occurring ARGs captured by Context-Seq included *aph(6)-Id, aph(3")-Ib, mphA, qnrS1, sul3*, and $bla_{CTX-M}$ among others. We also identified $bla_{TEM}$ co-occurring with disinfectant resistance (*qacEdelta1*) and mercuric reductase (*merA, merT, merC*) genes. Eight sequences containing $bla_{TEM}$ were greater than 18,000 bps and included multiple co-

occurring ARGs. For example, in the consensus sequences ≈ 20 kbp in the canine sample, $bla_{TEM}$, *sul2, aph(3")-Ib, aph(6)-Id, mrx, mphA* were identified across the 23,336 bp sequence. In the consensus sequence for the adult human fecal sample in household two, $bla_{TEM}$, *dfrA8, sul2, aph(3")-Ib, aph(6)-Id* occurred across the 19,716 bp sequence (Fig. 2C).

**ARGs were identified on plasmids and primarily in *E. coli*, *K. pneumoniae*, and *Hemophilus influenzae*.** Of the taxonomic identifications that were identified via Kraken2 and confirmed via BLASTN (Supplementary Data 1), $bla_{TEM}$ was identified on plasmids and chromosomes (consensus sequences not identified as plasmids with a plasX confidence <0.5) annotated as *K. pneumoniae*, *H. parainfluenzae*, *H. influenzae*, *E. coli* and Enterobacteriaceae (Fig. 3A). $Bla_{CTX-M}$ was identified on plasmids and chromosomes in *K. pneumoniae* and *E. coli* (Fig. 3B). Across hosts and genes ($bla_{TEM}$ and $bla_{CTX-M}$), the majority of sequences were identified as *E. coli* followed by *K. pneumoniae*. Other

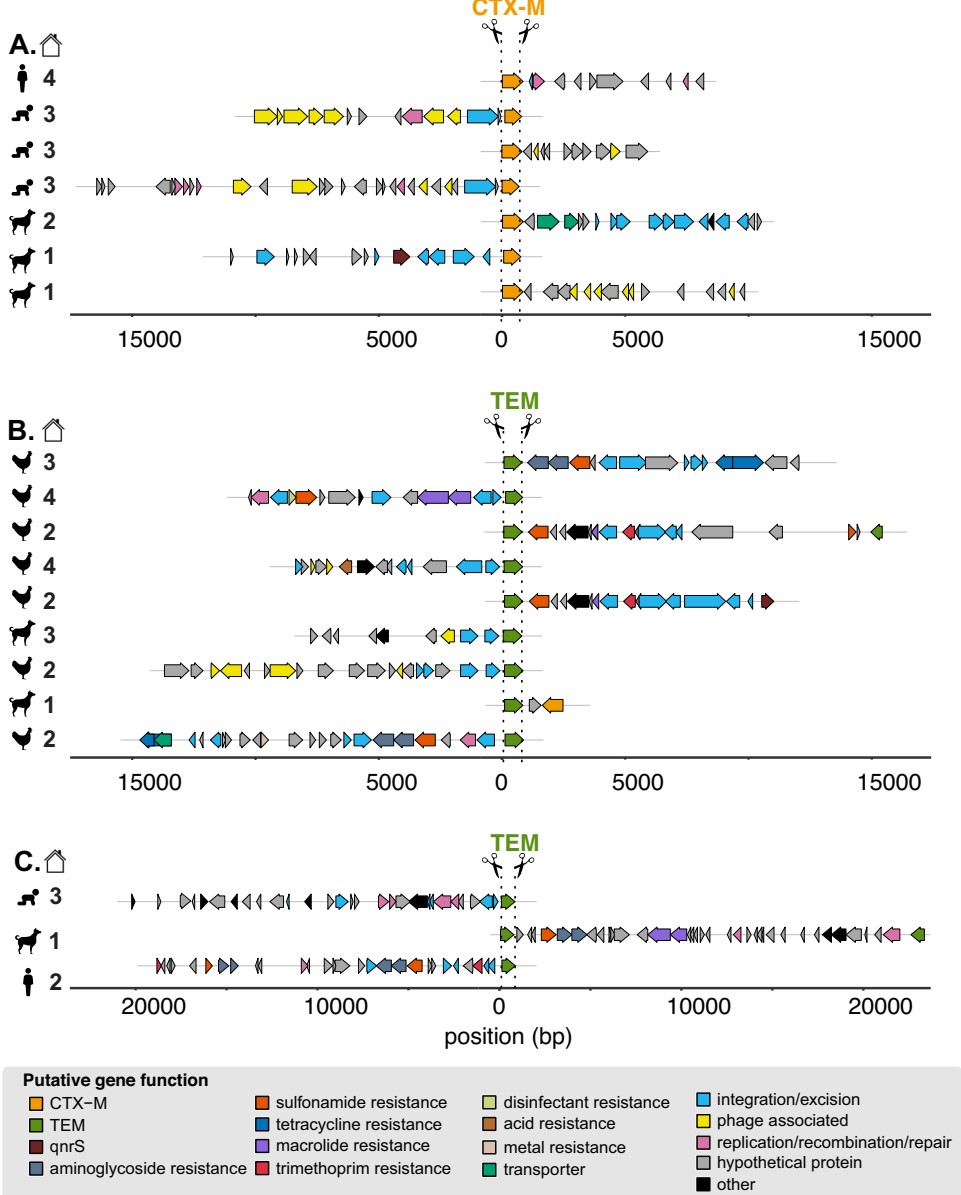

**Fig. 2 | Annotated enriched sequences containing *bla*$_{CTX-M}$ and *bla*$_{TEM}$ generated by Context-Seq.** A subset of sequences containing (**A**) *bla*$_{CTX-M}$, (**B**) *bla*$_{TEM}$, and (**C**) *bla*$_{TEM}$ ≈ 20,000 bp annotated for ARGs and mobile genetic elements. Sample type is indicated by symbol (child, adult, poultry, canine) and household by number. Note these sequences are a subset and some consensuses sequences were obtained from the same sample. In addition, all ARGs were aligned in the same orientation representing how cutting can proceed from the sense or anti-sense strand. Source data are provided in the Source Data file.

gammaproteobacteria, *H. parainfluenzae* and *H. influenzae*, were only found in human stool samples and not in animals.

We compared our method (Context-Seq in total DNA extracts) to a parallel study that cultured *E. coli* without antibiotics and sequenced up to five pooled isolates from the same samples[20]. We calculated the average coverage of the assembled contigs that resulted from Illumina sequencing and assembly of the cultured *E. coli* isolates. All instances of *bla*$_{TEM}$ and *bla*$_{CTX-M}$ identified in cultured *E. coli* by Illumina sequencing were also identified by Context-Seq in the same sample. In human fecal samples, the median coverage of the contigs was higher with Illumina sequencing compared to Context-Seq; coverage was 160x with Illumina sequencing (range: [17-735]) and 49 [7-1237] with Context-Seq (paired two-sided t-test p = 0.82). In animal samples, the median coverage with Context-Seq (171 [66-3852]) was greater than Illumina (138 [19-1099]), although not statistically significant (paired two-sided t-test p = 0.24) (Fig. 3C). In one poultry and one canine

sample, no contigs containing *bla*$_{TEM}$ or *bla*$_{CTX-M}$ were assembled from the cultured isolates, but both ARGs were identified in the canine and *bla*$_{TEM}$ was identified in the poultry using Context-Seq.

### ARGs are shared between human and animal hosts and across households

A total of 23 clusters ( > 80%ID over ≥3000 bp), one with *bla*$_{CTX-M}$ and 22 with *bla*$_{TEM}$, were shared between samples (Fig. 4A). 11 clusters were shared between humans and animals, 11 shared between animals only, and one cluster was shared between humans only (Fig. 4B). Of the animal-animal host sharing, seven clusters were observed in canines and poultry, one in canines only, and three in poultry only. 18 of the 23 clusters were found in more than one household and five were shared within individual households (Fig. 4B). Within the shared clusters, the *bla*$_{CTX-M}$ gene aligned to a group of *bla*$_{CTX-M}$ alleles highly similar in the target region (CTX-M-15/224/238/163/194/232). The *bla*$_{TEM}$ genes in

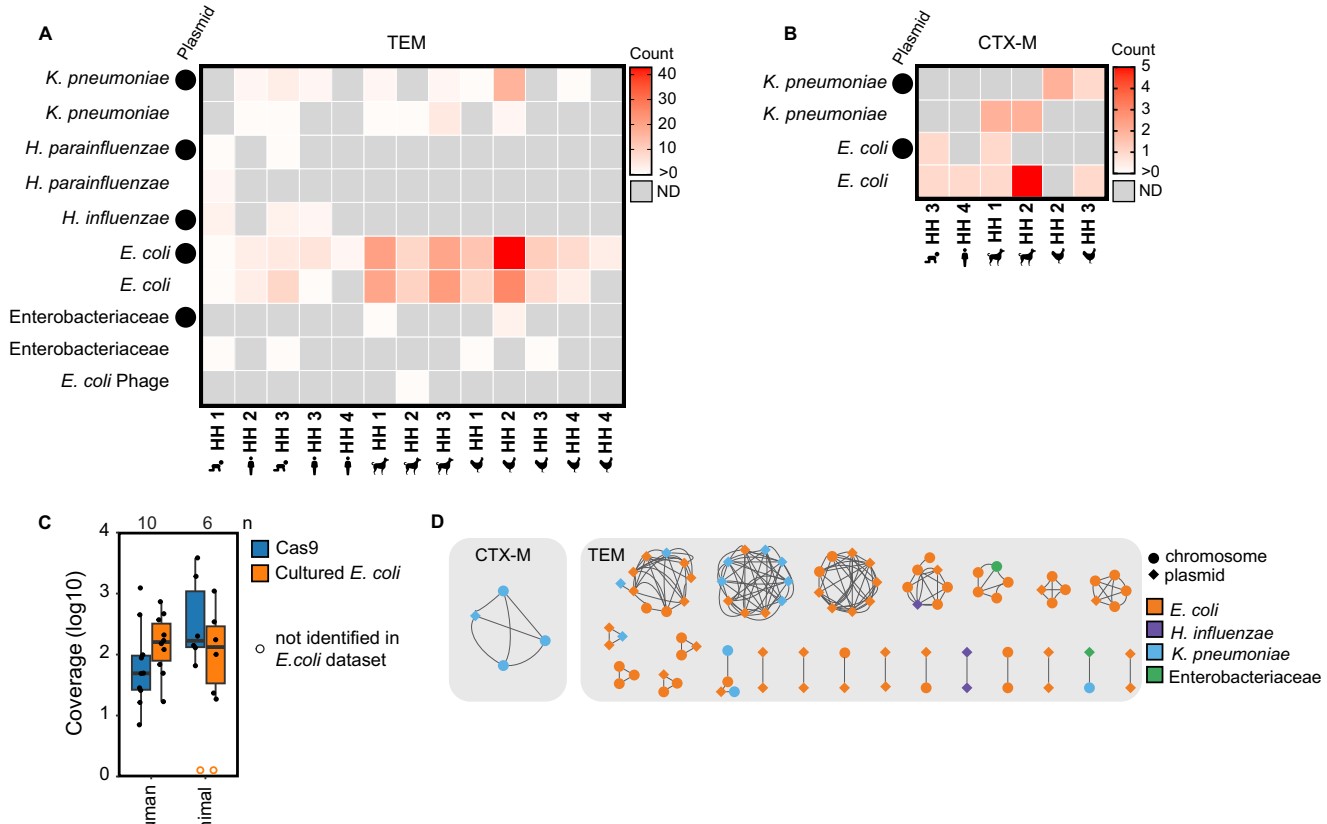

**Fig. 3 | Taxonomic identifications of Context-Seq sequences containing $bla_{TEM}$ and $bla_{CTX-M}$. A** The number of sequences containing $bla_{TEM}$ identified as each taxa by Kraken2 and confirmed by BLASTN. Plasmids (plasX ≥ 0.5) are indicated by a circle. **B** The number of sequences containing $bla_{CTX-M}$ identified as each taxa by Kraken2 and confirmed by BLASTN (supplemental data file 1). Plasmids (plasX ≥0.5) are indicated by a circle. **C** Comparison of Context-Seq to cultured and sequenced *E. coli* in the same sample. Coverage of the assembled contigs that resulted from 1.) the cultured *E. coli* using Illumina sequencing of the isolates and 2.) Context-Seq. Box shows interquartile range (25th to 75th percentiles) with the median and whiskers extending to 1.5 times the interquartile range. An open orange circle indicates that no $bla_{TEM}$ or $bla_{CTX-M}$ was identified in that sample using Illumina. **D** Taxonomic identification (color) and plasmid or chromosome designation (shape) in the 23 clusters ( > 80%ID over ≥3000 bp) that are shared between hosts (human, poultry, canine) and households. In some instances, more than one consensus sequence is shared between host and household. Taxonomy and plasmid are indicated for the highest percentage identity match. Source data are provided in the Source Data file.

the shared clusters aligned primarily to two groups of highly similar alleles (group 1: TEM-214/206/243/141/209/166 and group 2: TEM-217/234/104/198/228/135). Shared clusters generally contained multiple ARGs (e.g., *sul2*, *dfrA*, *tetA*, *aph(3")-Ib*) conferring resistance to sulfonamides, trimethoprim, tetracycline, and aminoglycosides in addition to the $bla_{TEM}$ or $bla_{CTX-M}$ targets. *TnpA*, which encodes for the transposase for transposon Tn3, was the most common integration/excision gene while *repA*, *repC*, and *parM* were the most common replication/recombination/repair genes.

Households one (HH 1) and two (HH 2) are located in Kibera while households three (HH 3) and four (HH 4) are located in Dagoretti South. Within the same community, households were ~0.1 miles apart while household in different communities were approximately 3 miles apart. 13 out of 23 clusters were shared between Kibera and Dagoretti South, five in Kibera only, and five in Dagoretti South only.

### ARGs were shared across hosts through plasmids and chromosomes

The $bla_{CTX-M}$ cluster was identified as *K. pneumoniae*, shared between three households, and shared between canines and poultry (Fig. 3D and Fig. 4B). The sequence containing the $bla_{CTX-M}$ cluster was identified as a likely plasmid in one of the four samples (Fig. 3D). $Bla_{TEM}$ was shared across households and hosts on consensus sequences identified as *E. coli*, *K. pneumonia*, and *H. influenzae* (Fig. 3D). Of the shared $bla_{TEM}$ clusters, approximately half were shared on the same element (plasmid to plasmid or chromosome to chromosome) and half were shared between elements (plasmids and chromosomes). The sole human to human shared cluster (cluster 3 between children in HH 1 and 3) was classified as a *H. influenzae* plasmid in both samples.

### Discussion

Here we developed Context-Seq using Cas9 targeted sequencing paired with long-read sequencing to enrich for fragments of DNA containing two clinically relevant ARGs, $bla_{TEM}$ and $bla_{CTX-M}$. We identified sharing of ARGs and genomic context between humans and animals, as well as between poultry and canines in urban Kenyan households, emphasizing the importance of the One Health approach to combatting AMR. Suspected hosts of ARGs were not just limited to *E. coli* but also included *K. pneumonia*, *H. influenzae*, and *H. parainfluenzae*. *E. coli* remains one of the most commonly characterized antimicrobial-resistant organism[36], but many others are important to investigate in the context of AMR transmission.

An important finding of this work is the occurrence of non-*E. coli* hosts, primarily *K. pneumoniae* and *H. influenzae*. *K pneumoniae* is a leading cause of antimicrobial resistant infections[37,38], especially in hospitalized patients[39] and is on the WHO global priority pathogens list[40]. While *K. pneumoniae* can survive in multiple environments including soil, water, and the intestinal tract of humans and animals[39],

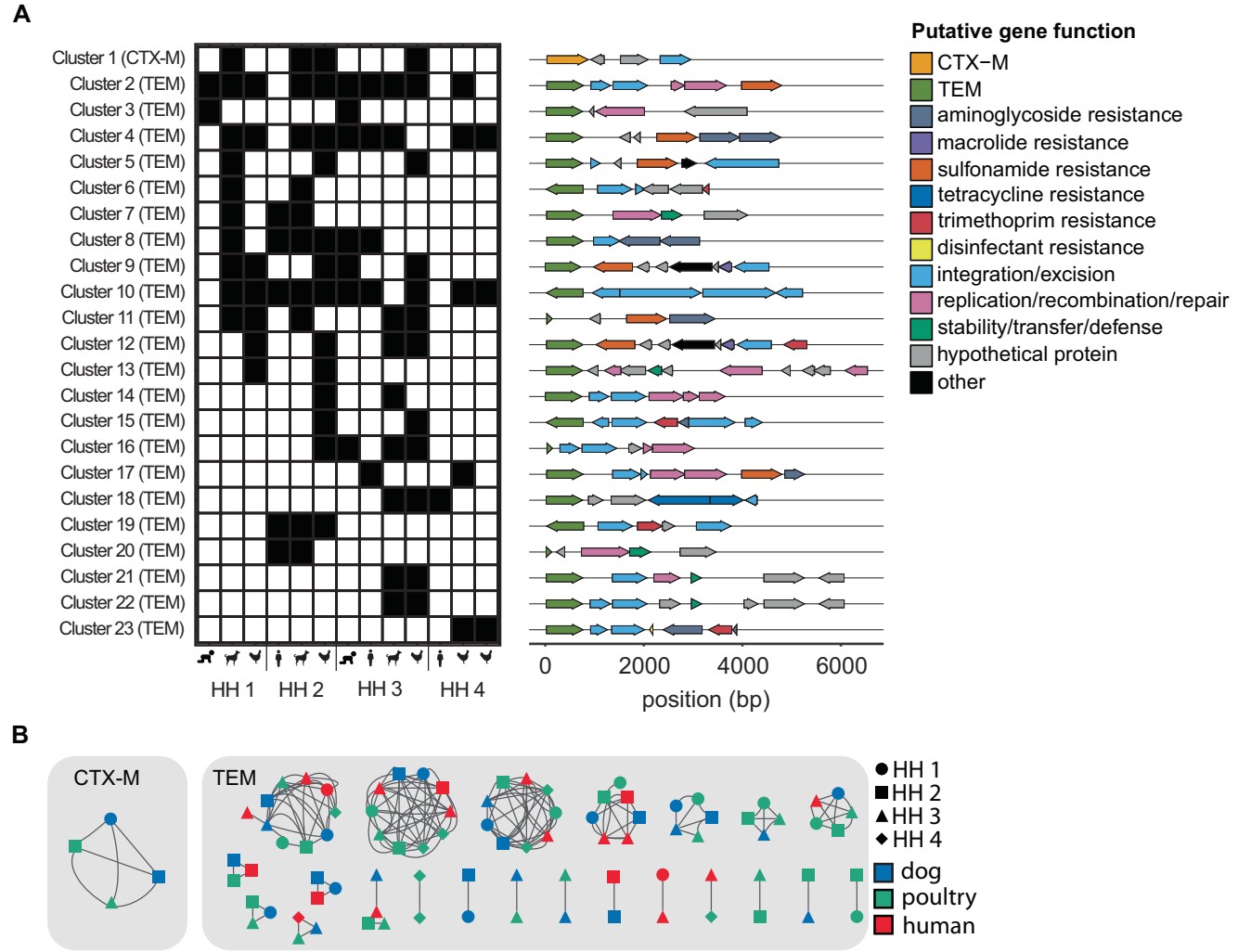

**Fig. 4 | 23 clusters containing $bla_{TEM}$ or $bla_{CTX-M}$ were shared between samples.** **A** Presence (black)/absence of shared clusters ( > 80%ID over ≥3000 bp) by sample type and household. ARG and mobile genetic element annotation of shared clusters. **B** Shared clusters annotated by household (shape) and host type (color). Source data are provided in the Source Data file.

few studies have investigated resistant *K. pneumoniae* from a molecular epidemiology, One Health perspective (i.e., strain sharing in humans, animals, and the environment). A previous study in Kenya collected *K. pneumoniae* from community fecal samples, healthcare-associated fecal samples, and hospital surfaces across multiple counties. Among extended spectrum beta-lactamase (ESBL) isolates, $bla_{CTX-M-15}$ and $bla_{TEM-181}$ were the most common genes[41]. We identified $bla_{CTX-M-15\ like}$ in *K. pneumoniae* in a shared cluster, while $bla_{tem-181\ like}$ was identified in one sequence, but not in a shared cluster. Most $bla_{TEM}$ genes were highly similar to non-ESBL genes $bla_{TEM-141}$ and $bla_{TEM-135}$ (groups 1 and 2 above). Very few studies have been conducted on *K. pneumoniae* in household animals, although one study investigated *K. pneumoniae* isolated from raw meat samples in Nairobi[42]. They found high resistance to ampicillin but very few isolates had ESBL resistance[42]. Further, *H. influenzae* is a leading cause of bacterial respiratory infection[43] with recent increasing resistance to beta-lactams[44]. Compared to *K. pneumoniae* and *E. coli*, there are even fewer studies of resistant *H. influenzae*, especially in LMICs. One study investigated resistant *H. influenzae* in Morocco and found one-third of isolates carried resistance genes to beta-lactams[45]. Most were susceptible to ESBLs, primarily demonstrating resistance to ampicillin and amoxicillin[45].

Approximately half of shared ARGs and their genomic context (11/23 clusters) were shared between animals. An extensive previous study

of *E. coli* across hosts in Nairobi (the UrbanZoo project) found highly similar resistomes among livestock poultry, both within and between households[11]. This work hypothesized the similar poultry resistomes were the result of similar antimicrobial selective pressure since use of antimicrobials for therapeutic or prophylactic purposes is consistent across Nairobi[46]. We observed significant genetic synteny between poultry and canines (7 out of 11 animal-animal clusters), however the UrbanZoo project did not investigate canines. A similar mechanism may exist for canines and poultry in that they may consume similar antibiotics, and their guts could select for similar ARGs[47]. Another potential mechanism of AMR acquisition in canines is through scavenging. Previous work has demonstrated that the widespread waste (including human and animal feces as well as garbage) across the urban landscape of Nairobi can serve as a reservoir for AMR[48]. Canines could acquire similar AMR as poultry through eating of poultry feces.

Human and animal overlap (or high genetic similarity) was observed in the other half of shared clusters (11 out of 23). Previous evidence for human and animal resistome sharing has been mixed[10,12,49,50]. In a related study, where *E. coli* was isolated from humans, animals, and the environment in the same households as our work, human and animal strain sharing was rare[20]. The majority of *E. coli* strain similarity was observed between humans and stored drinking water, and poultry and soil, implicating the environment as a reservoir between hosts[20]. The paired study also identified that within

household sharing was more common than between household sharing, yet we found that over half of sharing events occurred in both communities. The observed higher relative degree of sharing between humans and animals and between communities in our work is likely due to method specific differences. Here, we investigated ARG containing DNA fragments only and our metagenomic-based approach captured additional species outside of *E. coli*. We did not apply enrichment sequencing to soil and water, thus cannot compare findings related to the environment. Together, these paired efforts highlight that multiple approaches may be needed to obtain a more complete understanding of AMR in a given context. Finally, our results are consistent with the paired study and UrbanZoo[11] in that when human to animal overlap is observed, it may occur between households.

We observed similar ARGs and surrounding genomic context in both plasmids and chromosomes. We note that *chromosome* was classified as sequences that were not identified as plasmids, and not through whole genome or strain analysis. Diverse mobile elements carrying ARGs have been observed in humans and animals. Horizontal gene transfer can facilitate transfer of AMR through these reservoirs. Previous work on *E. coli* in Nairobi concluded organismal spread, rather than transduction or transformation, was the dominant mechanism of highly similar mobile elements between human and animal hosts[48]. Similarly, the paired study of *E. coli* isolates in our same study households found strain sharing was more likely to contribute to resistome sharing than horizontal gene transfer[20]. Notably both studies were conducted on a single species, and our work demonstrates shared ARGs and genomic context between species (*E. coli* and *H. influenzae*, *K. pneumoniae* and *E. coli*) which likely occurred through horizontal gene transfer.

Cas9-targeted sequencing is a promising method for target specific enrichment. For all samples in which $bla_{CTX-M-15}$ or $bla_{TEM}$ was found in sequenced *E. coli* isolates, the ARG was also detected using Context-Seq. Our target ARGs were identified in two samples with Context-Seq but not by culture, likely because the selective conditions of culturing enabled other strains to outcompete the ARG-containing bacteria. Context-Seq also resulted in a significant increase in coverage of our target genes compared to untargeted sequencing, though performance varied by sample. While the on-target read percentage was relatively low, the overall coverage of target genes was still high. The background DNA is likely the result of incomplete enzymatic reactions during library preparation. For example, a high dephosphorylation efficiency, even as high as 99.9%, would still result in millions of phosphorylated DNA fragments given our high input DNA concentration. Some of the most significant factors for enrichment are likely initial target concentration, guides available per target, and pores available per sample. For example, a previous study investigated the detection of *K. pneumoniae*-associated genes, including $bla_{CTX-M-15}$, using fecal samples spiked with variable colony forming units of *K. pneumoniae*[17]. The authors did not identify any $bla_{CTX-M-15}$ reads in the untargeted samples but did identify 1-8 reads in the targeted, whereas we identified up to 419 reads as $bla_{CTX-M-15}$. The previous work used a guide pool of 31 pairs of guides targeting many *K. pneumoniae* genes, while we used two guide pairs (one for each gene)[17]. This reduces the guides targeting $bla_{CTX-M-15}$ whereas half the guides in our pool targeted $bla_{CTX-M-15}$ genes. They also multiplexed samples, which is more cost effective but reduced the available pores and sequencing effort per sample. This previous work did demonstrate high coverage across *K. pneumoniae* genes[17], and together with our study, demonstrates the utility of Cas9-targeted sequencing.

This work has several limitations. Oxford Nanopore Technologies' long-read sequencing is a relatively new technology and is consistently changing to improve the nominally high error rate (≈90–95% accuracy). This project was conducted on previous generation flow cells (R9.4.1), which are available from ONT upon request. Additional

validation would need to be conducted for R10.4.1 since the duplex chemistry is a significant shift from the previous versions. Similar methods could be applied to alternative long-read technology such as PacBio[51]. In addition, taxonomic identification of ARG hosts is challenging due to plasmid sharing, especially in Enterobacteriaceae[52]. We only included identifications that were consistent across methods but in many cases reflects a possible host. Also, due to the high sequence similarity of beta-lactamase alleles and relatively high error rate of nanopore, it is possible allelic variation in our target ARGs is not captured in consensus sequences. Finally, we did not process environmental samples in this study; future work to process environmental samples (e.g. soil and water) with Context-Seq is recommended for a complete One Health approach to investigating AMR[53]. The feasibility of environmental matrices also varies as the input DNA requirement (1.5-3.0 µg) would require extensive concentration for some sample types (e.g. surface or drinking water).

Additional modifications could be made to improve Context-Seq for future applications. While we demonstrate the utility of this assay with two ARGs, there is potential for ARG multiplexing, sample multiplexing, and further optimization of the enriched alleles. A previously published method (FLASH) used Cas9 with short-read sequencing to target detection of 127 genes with 5513 guides[14]. Since many ARGs are co-located[54], a guide pool of this size would likely be counterproductive to obtaining long reads but there is possible room for target expansion before compromising read length. One potential area of expansion is including guides for different alleles (e.g. $bla_{CTX-M\ Group\ 1}$ vs. $bla_{CTX-M-15\ Group\ 9}$) as they would likely be present on different DNA fragments. Further, the greatest potential for improving this method is sample multiplexing[17] to reduce costs. While we ran each sample on a single flow cell, multiplexing on a MinION or promethION would significantly reduce the per sample cost. However, multiplexing is non-trivial and requires careful optimization to reduce off-target reads as it adds an additional step where non-target fragments can shear and introduce phosphorylated ends available for adapter ligation. Lower cost Context-Seq could be transformative to inform transmission dynamics of AMR through human, animal, and environmental reservoirs in diverse settings.

## Methods

The study was approved by the Kenya Medical Research Institute (KEMRI) Scientific and Ethics Review Unit (Protocol number 3823) and the Tuft Health Sciences Institutional Review Board (13205). Additionally, a research permit was granted by the Kenyan National Commission for Science, Technology, and Innovation. Written informed consent was obtained from each adult participant. For children, both child assent and parental written consent were obtained.

### Sample collection and processing

Poultry-owning households from Dagoretti South and Kibera subcounties of Nairobi, Kenya were sampled in June-August 2019. Up to three poultry fecal samples and one canine fecal sample were collected during an initial visit. To collect animal feces, a sterile plastic scoop was used to transfer feces from the top, center layer of a fresh fecal pile. Approximately one week after the first visit, households were revisited to collect human stool from one household member in the following three age groups: child aged 0–4 years, child aged 5–14 years, and adult aged 15 years or older. A stool collection kit was provided during the first visit, which included a 50 mL plastic pot with a sterile scoop for each member with instructions on how to collect the sample. The primary caretaker of each household was informed by mobile phone one day prior to the revisit to collect stool from the previous night or the morning of the revisit day. All human and animal fecal samples were placed in a cooler filled with ice and transported to KEMRI. 1 g of fresh feces was aliquoted for storage at −80 °C without preservatives. DNA was extracted from animal samples at KEMRI and stored at −80 °C

until transport. Human fecal samples and DNA extracts from animals were shipped to Tufts University on dry ice. For all fecal samples, DNA was extracted from 0.2 g of feces using Qiagen's Powersoil Pro kit according to the manufacturer's instructions.

## Statistics & reproducibility

No statistical method was used to predetermine sample size. Samples were not randomly selected but chosen based on nucleic acid yields and availability of all sample types from households. The sex, based on reporting, for all sequenced human stool samples is available in Supplementary Table 2. Sample size was too small for disaggregated analyzes.

## Taqman array card

Seven adult fecal samples, 13 child fecal samples, four canine, and 15 poultry samples from eight households were prescreened for ARGs[55] using a Taqman Array Card prior to enrichment sequencing. 22 targets were run in duplicate, 14 of which were ARGs. Samples positive for CTX-M group 1 and the TEM assay were candidates for enrichment sequencing (Supplementary Table 2) (see supplementary information for more details).

## gRNA Design

Candidate guide RNAs (gRNAs) for ARGs were determined by CHOPCHOP[34]. A custom program (https://github.com/Shruteek/Optimized-sgRNA-Design) was created to screen off-target effects in representative metagenomes. The program, implemented in Python, used empirical methods for on- and off-target effect analysis by taking in a candidate sequence and a sample metagenome, and returning a heuristic representing the overall likelihood of the candidate sequence to experience off-target effects in the metagenome. Based on guide RNA binding behavior[35], we counted off-target sites as valid only if they had 5 or fewer mismatches, 1 or fewer mismatches in the 10 PAM(protospacer adjacent motif)-proximal base-pairs, and a PAM of the form 5'-NGG-3' or 5'-NRG-3'. Bowtie[56] was used to identify potential off-target sites, then each off-target site was evaluated for its binding likelihood based on the number of PAM sequences in the forward and reverse sequence[57], the 1- and 2-base-pair nucleotide features in and around the site[58], the identity of each nucleotide[58–60], the individual nucleotide mismatches between the site and the guide[35], and the proximity of each mismatch to the PAM[35,61,62]. The binding likelihood scores of the on-target sequence and each off-target site were normalized from 0 to 100, the latter corresponding to maximum binding odds for a perfectly stable matching sequence, and all off-target likelihood scores for a single guide were summed and subtracted from the on-target likelihood score to generate the heuristic for off-target effects.

## Library preparation

We modified a previously published Cas9 enrichment protocol[15,63]. To evaluate performance and test modifications, we made a model system comprised of an E. coli isolate with $bla_{CTX-M-55}$ and $bla_{TEM-1}$ genes spiked into composited DNA extracted from Kenyan soil (see supplementary information for additional details). Soil was chosen as the test matrix to confirm the method works on a complex sample type. Unless otherwise specified, the protocol was performed as described below. We primarily evaluated modifications for only CTX-M unless the modification was specific to multi-target detection. We made the following adjustments 1.) For adaptive sequencing, the MinKNOW software was set up in adaptive mode using the $bla_{CTX-M-55}$ gene as the reference and aligning up to 200 bps. Adaptive sequencing is a software-based method that allows the MinKNOW software to read the first few hundred base pairs of a fragment and selectively reject the fragment from the pore if it is classified as off-target[64]. 2.) For longer Cas9 cut time, Cas9 digestion proceeded for 2 h instead of 20 minutes.

Cas9 cut time is a balance of increased time for on-target binding and cutting but may increase off-target binding. 3.) For two guides per target per strand (sense and antisense), guides were added in an equimolar mix of 0.75 μL each to a 0.5 mL centrifuge tube and 1uL of the mix was complexed with Cas9. Multiple Cas9 guides are used for one region of interest in similar approaches[14,15]. The protocol below describes the addition of Proteinase K as that was incorporated in our final procedure. Proteinase K was added to remove Cas9 with the hypothesis that Cas9 could block pores leading to early pore death.

CrRNAs (CRISPR RNA) and tracrRNAs (trans-activating CRISPR RNA), together forming the gRNA, were resuspended to a final concentration of 100 μM in duplex buffer (IDT). 8 μL of nuclease free water, 1 μL of tracR, and 1 μL of crRNA were mixed and heated at 95 °C for 5 minutes for duplex formation. To create the ribonucleoprotein complex (RNP), 1X CutSmart Buffer (NEB), 2 μM of gRNA, 0.5 μM of HiFi Cas9 Nuclease V3 (IDT), and nuclease free water were combined to a total reaction volume of 30 μL. The reaction was incubated at room temperature for 20 minutes. Input DNA (~1.5–3.0 μg) was dephosphorylated in a 60 μL reaction composed of 6 μL 1X CutSmart buffer, DNA, nuclease free water, and 3 μL of QuickCIP (NEB). The reaction was incubated at 27 °C for 20 minutes followed by inactivation at 80 °C for 2 minutes. Dephosphorylation of the sample DNA is needed so only the phosphorylated ends of the DNA that result from Cas9 cutting are ligated to sequencing adapters. For Cas9 cleavage and A-tailing, dephosphorylated DNA was split up into two reactions (2 reactions of 30 μL). Input DNA was split to allow for Cas9 cutting on the sense and antisense strands separately using two sets of guide RNAs for the two target ARGs. Cas9 cutting introduces blunt end cuts and A-tailing is used to create TA overhangs for adapter ligation. 30 μL of DNA and 10 μL of RNP were mixed and incubated at 37 °C for 15 minutes. 1 μL thermolabile Proteinase K (NEB) was added and incubate at 37 °C for 10 min. Proteinase K was then heat inactivated at 65 °C for 10 minutes. 1 μL dATP (Invitrogen) and 1 μL Taq polymerase (NEB) were added to the mixture and incubated at 37 °C for 15 min followed by 72 °C for 5 minutes. To ligate on sequencing adapters, ligation mix was prepared by adding 9 μL nuclease free water, 40 μL ligation buffer, 20 μL T4 quick ligase (NEB), and 7 μL of adapters. 38 μL of the adapter mix was added to each reaction and incubated at room temperature for 10 minutes on a tube rotator. Equal volume of TE buffer was added to each reaction and then the two libraries (one for sense and another for antisense strand) were pooled. 0.5X Ampure beads (Beckman Colter) were added. The reaction was rotated for 5 minutes followed by incubation at room temperature on a bench top for 5 minutes. Magnetic beads were washed with 250 μL of long fragment buffer. After addition of 13 μL of elution buffer, beads were incubated at 37 °C for 30 minutes. MinION flow cells were loaded according to the manufacturer's instructions with 12 μL DNA in elution buffer, 25.5 μL loading beads, and 37.5 μL sequencing buffer.

## Sequencing

Samples were run on MinION (FLO-MIN106) R9.4.1 flow cells using a MK1B sequencer for 72 h or until there were no available pores. Runs were operated using MinKNOW software (v22.05.5, v22.10.10, v22.12.7, v23.04.6).

## Data analysis

Fast5 files were basecalled using guppy (v6.1.5, v6.3.9, v6.4.6, v6.5.7) with a minimum quality score of 7. Porechop (v0.2.4)[65] was used to trim remaining nanopore sequencing adapters. Usearch (v11.0.667)[66] was used to sort trimmed reads by length (-sorbylength -minlength 500) and cluster reads with a minimum overlap of 1500 bp at 85% identity (-cluster_fast -id 0.85 -strand both -mincols 1500). Three cycles of racon (v1.4.20)[56] (racon using minimap2 for overlaps) followed by medaka (0.11.5)[67] (medaka_consensus) were used to polish centroids with the reads assigned to the same cluster, generating one consensus

sequence per cluster. Singletons were included without polishing. Antibiotic resistance genes (ARGs) were identified in resulting consensus sequences (including singletons) using Minimap2 (2.22-r1101)[68] to map against the Comprehensive Antibiotic Resistance Database (CARD) (v3.2.6) (-cx map-ont)[69]. Mobile genetic elements (MGEs) were annotated using the mobileOG database (beatrix-1.6) (./mobileOGs-pl-kyanite.sh -k 15 -e 1e-20 -p 90 -q 90). The mobileOG[70] database is a manual curation of MGEs from ICEBerg, ACLAME, GutPhage Database, Prokaryotic viral orthologous groups, COMPASS, NCBI Plasmid RefSeq, immedb, and ISfinder, along with homologs of the manually curated sequences. Consensus sequences were also annotated using Prokka (v1.14.5)[71]. Taxonomy was assigned to consensus sequences using Kraken2 (v2.0.7-beta)[72] (kraken2 -threads 24) with the default full database. BLASTN was used to map consensus sequences against NCBI core non-redundant nucleic acid database (core_nt). Kraken2 names were retained if the same taxon was identified in at least one of the top five BLASTN matches. If there was no agreement between Kraken2 and the top five BLASTN hits, the consensus sequence was not included in the taxonomy analysis (supplemental data file 1). PlasX[73] was used with anvio created annotations (anvi-gen-contigs-database, anvi-export-gene-calls, anvi-run-ncbi-cogs, anvi-run-pfams, anvi-export-functions) to identify the probability contigs were plasmid sequences(plasx search_de_novo_families –splits 32 –threads 128). Sequences with a PlasX score > 0.5 (0 likely not plasmid, 1 likely plasmid) were labeled as plasmids. BLASTN (2.12.0) all-versus-all was used to identify regions of genetic synteny between samples at greater than 80% identity for 3000 bps (-perc_identity 80 -outfmt 6). Annotated consensus sequences were visualized in R (4.3.2) using ggenes (0.5.1). Coverage plots for benchmarking were visualized using genomicRanges (1.54.1) and genomicAlignments (1.38.0).

### Reporting summary

Further information on research design is available in the Nature Portfolio Reporting Summary linked to this article.

## Data availability

All fastq files generated in this study have been deposited in the Sequence Read Archives under BioProject PRJNA1157857. Source data are provided with this manuscript.

## Code availability

All codes for the guide design tool are available at https://github.com/Shruteek/Optimized-sgRNA-Design, and a frozen version used in this manuscript is available here [https://doi.org/10.5281/zenodo.15198795].

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

## Acknowledgements

This work was supported by grant OPP1129535 from the Bill and Melinda Gates Foundation, the Chan Zuckerberg Biohub, San Francisco, and the National Science Foundation Award 2143622. Support also came from the National Center for Advancing Translational Sciences, National Institutes of Health, Award Number UL1TR002544 and the Stuart B. Levy Center for Integrated Management of Antimicrobial Resistance at Tufts (Levy CIMAR), a collaboration of Tufts Medical Center and the Tufts University Office of the Vice Provost for Research (OVPR) Research and Scholarship Strategic Plan (RSSP). The NSF Postdoctoral Research Fellowships in Biology Program under Grant No. 1906957 supported ERF. We thank Maya Nadimpalli, Robert Gilman, and Monica Pajeulo for their contribution of the positive control isolate (NIH R01AI108695-01A1 and Tufts Springboard award). We thank Honey Mekonen, Ritwicq Arjyal, Joana Cabrera, and Andres Dextre for research assistance. Any opinions, findings, and conclusions or recommendations expressed in this material are those of the author(s) and do not necessarily reflect the views of the funding organizations.

## Author contributions

Project conceptualization was performed by E.R.F. and A.J.P. Methodology for this work was developed by E.R.F., S.K., S.A.M., A.P. Samples were collected by J.M.S. and B.C. Laboratory experiments were performed by E.R.F., S.K., C.M. and A.P. Data analysis was conducted by E.R.F., S.A.M., S.K. and C.M. Visualization of data and results was performed by E.R.F. Funding for this work was acquired by E.R.F., A.J.P. and S.M.N. Writing of original draft was carried out by E.R.F., S.A.M. and A.J.P. Reviewing and editing of the manuscript was performed by all.

## Competing interests

The authors declare no competing interests.
