## [Transparent Peer Review file · Nature Communications]

Context-Seq: CRISPR-Cas9 Targeted Nanopore Sequencing for Transmission Dynamics of Antimicrobial Resistance

Corresponding Author: Professor Amy Pickering

Version 0:

Reviewer comments:

Reviewer #2

(Remarks to the Author)

The manuscript by Fuhrmeister et al. describes the development and application of Cas9-mediated targeted metagenomics for investigating transmission dynamics of ARGs across One Health compartments. The technique is creative and well documented for reproducibility, and the example dataset in Kenya provides excellent insights into cross-compartment gene sharing networks. There are a few areas where limitations to the assay as executed could be better addressed. I have a few comments to aid in the presentation, clarity, and impact of the work.

Major Comments/Questions:

- Clustering and dereplication of reads at 85% to generate polished, consensus sequences is a good strategy for minimizing the ONT error rate, but it seems to limit the impetus of the analysis. ARGs in the beta-lactam family are highly similar, up to singular amino acid substitutions. Has clustering undermined the specificity of the assay? For example, Figure 2 illustrates overlapping synteny but does not denote the specific alleles. Were there any exact ARG-MGE combinations that occurred across samples to illustrate more definitive evidence of gene sharing?

- I don't understand how so little reads would align to your targets given Cas9-selection and adaptive sequencing (Lines 148-154, Table S7). How is there so much background noise in the data and what are the off-targets? Also, why would it be that there'd be such a narrow range of read lengths if it's sequencing unrestrictedly in each direction?

- In general, the methods read as a very detailed step-by-step protocol, but it seems to be missing the "why" for some steps. Consider reworking the methods (where possible) to be more generalized with a detailed, step-by-step protocol that can be published separately on protocols.io. For example, if there is no deviation from the ONT ligation sequencing kit used from lines 411-420, then you can simplify the explanation. Many decisions were obviously made to troubleshoot and optimize this assay, and I think the more conversational information that is included with justifications, the better. Could have missed it, but a reasoning/purpose for the addition of Proteinase K is needed.

- For the adaptive sequencing, it reads as if only CTX-M-55 is used as a guide to capture both CTX-M and TEM variants.

Please clarify.

Minor Comments/Suggestions:

Line 76: "overlap in AMR elements" is too vague. Consider using the concept of "genetic synteny" to describe this work throughout.

Line 77-78: *Captures a small fraction of the population of a given species.

Is there ever a comparison to the sensitivity between the TaqMan assay and Context-Seq?

Line 135: Is it accurate to describe this a "mock community"? How can you be sure that those ESBLs weren't already present? Also, it reads as if whole cells were spiked into raw DNA extracts which threw me off before I read the supplemental.

Line 205-206: Denote coverage as "160x".

Line 303-304: This is very interesting. Were the sampled households in linked communities/compounds to encourage the inferred sharing? It may be worth noting the relative proximity of the sample households to one another (if I've missed that in text).

Line 305: When the phrase "ARG sharing" is used here it implies HGT, although cross-cell HGT is not typical via chromosomal ARGs. Maybe rephrase to say corresponding genetic contexts were observed between chromosomes and plasmids across samples?

315-324: Given the exorbitant DNA requirements for ONT (1.5-3 ug), many environmental matrices, notably surface, ground, and drinking water, may be limited by the technique (notwithstanding newer library prep kit requirements).

Need to define PAM, CrRNA, and tracrRNA.

423-424: How long were the flow cells run?
Bioinformatics: Give parameters/settings used for each software.

(Remarks on code availability)

The code provides sufficient information for reproducibility but I did not attempt to install and test.

Reviewer #3

(Remarks to the Author)

Review of NCOMMS-24-62079-T

Fuhrmeister et al. provides an example of how targeted Nanopore sequencing with Cas9-guided adapter ligation can be used to characterise selected antimicrobial resistance genes within their genomic context. The authors demonstrate that chromosomally and plasmid-encoded blaTEM and blaCTX-M genes are shared between adults, children, poultry, and dogs in animal-owning households in Nairobi, Kenya. The manuscript is clearly presented and written.

I have a number of issues for the authors to consider:

1) The idea to use Nanopore Cas9-targeted sequencing to characterise antimicrobial resistance genes within their genomic context is not entirely new. For example, Cottingham et al. used the same method to screen for antimicrobial resistance genes (as well as other targets) in three human faecal samples spiked with *Klebsiella pneumoniae* at varying abundance (DOI: 10.1128/msystems.01413-24). The authors should mention this study (as well as other similar studies if they exist) and compare the methodologies and results.

2) The authors selected two targets (blaTEM and blaCTX-M) and screened 13 faecal human, canine, and avian samples collected from four households that were positive for blaCTX-M group 1 and blaTEM. Despite the small sample size, the authors were able to identify genetically distinct clusters containing blaTEM and one cluster containing blaCTX-M that were shared within and between households. While humans and animals in close contact are already known to share bacteria and antimicrobial resistance genes, the study clearly demonstrates the usefulness and advantages of Nanopore Cas9-targeted sequencing compared to other methods. The authors also screened a mock community comprising an *Escherichia coli* isolate with blaCTX-M-55 and blaTEM-1 genes spiked into a composite sample of extracted DNA from Kenyan soil, but these results are not clearly presented. As the authors mention, it would have been interesting and very relevant in a One Health perspective to investigate whether Nanopore Cas9-targeted sequencing can be used as a tool to screen for antimicrobial resistance elements in soil and wastewater samples. This could be done by spiking soil and wastewater samples with the *Escherichia coli* isolate containing the blaCTX-M-55 and blaTEM-1 genes or by screening pristine soil and wastewater samples that are known to be positive for blaTEM and blaCTX-M (or other clinically relevant targets). This information would significantly increase the novelty and impact of the manuscript.

3) Please explain what you mean by "clusters" in l. 39-40 and throughout the manuscript.

4) It is unclear to me what "pairs" refers to in l. 125.

5) The sentence in l. 144-146 is unclear and should be rephrased.

6) What does "(35)" in l. 153 refer to? Is it the standard deviation? In that case, it is very high.

7) Please explain your findings in l. 209-211. Is it because blaTEM and blaCTX-M were located in another bacterial species than *Escherichia coli*?

8) Chromosomal and plasmid sequences were screened for taxonomic associations by Kraken2 and BLASTN. Please state whether all sequences were associated with a single bacterial species and discuss the potential confounding effect of interspecies horizontal gene transfer.

(Remarks on code availability)

Version 1:

Reviewer comments:

Reviewer #2

(Remarks to the Author)

The authors have made substantial improvements to the clarity and justification of the manuscript and all methods. After review, I could only find one typo:

Line 448 of Track Changes Doc: *leading to early pore death*

(Remarks on code availability)

The code is appropriately available and sufficient instructions are provided for testing and reproducibility.

Reviewer #3

(Remarks to the Author)

Review of NCOMMS-24-62079A

The authors have addressed all my issues and I am satisfied with the revision.

(Remarks on code availability)

REVIEWER COMMENTS

Reviewer #2

The manuscript by Fuhrmeister et al. describes the development and application of Cas9-mediated targeted metagenomics for investigating transmission dynamics of ARGs across One Health compartments. The technique is creative and well documented for reproducibility, and the example dataset in Kenya provides excellent insights into cross-compartment gene sharing networks. There are a few areas where limitations to the assay as executed could be better addressed. I have a few comments to aid in the presentation, clarity, and impact of the work.

Major Comments/Questions:

1. Clustering and dereplication of reads at 85% to generate polished, consensus sequences is a good strategy for minimizing the ONT error rate, but it seems to limit the impetus of the analysis. ARGs in the beta-lactam family are highly similar, up to singular amino acid substitutions. Has clustering undermined the specificity of the assay? For example, Figure 2 illustrates overlapping synteny but does not denote the specific alleles. Were there any exact ARG-MGE combinations that occurred across samples to illustrate more definitive evidence of gene sharing?

- We thank the reviewer for bringing up this point. We agree that clustering adds complications for beta-lactam genes that are highly similar. Given the per-read error rate of nanopore sequencing, clustering is used to improve sequencing accuracy of the genomic context but highly similar alleles can indeed be missed. We have included mentions of this in the discussion.
- With respect to genomic synteny, while we set a threshold of 80% the average is around 97% so our confidence in the 3000 bp overlapping regions in Figures 3D and 4B is high.
- In order to discuss the beta-lactam alleles, there is the limitation described by the reviewer with clustering but our guides also do not capture the full length sequence. Short regions are missed at the start and end of the reads if the sense and antisense strands are not clustered together. We addressed this by describing alleles by their allele groups in the text. For example, a consensus sequence that aligns equally well to TEM 214, 206, 243, 141, 209, and 166 is described as TEM-214/206/243/141/209/166.

Revised text as it appears in text lines 395-397: “Also, due to the high sequence similarity of beta-lactam alleles and relatively high error rate of nanopore, it is possible allelic variation in our target ARGs is not captured in consensus sequences.”

2. I don't understand how so little reads would align to your targets given Cas9-selection and adaptive sequencing (Lines 148-154, Table S7). How is there so much background noise in the data and what are the off-targets? Also, why would it be that there'd be such a narrow range of read lengths if it's sequencing unrestrictedly in each direction?

- An analysis of the reads that do not align to an ARG is consistent with the expected taxonomy of human and avian fecal samples. The specificity of this assay is only tied to free 5'-phosphate ends, which we try to generate selectively by Cas9 cleavage but they can also arise from other sources – including incomplete dephosphorylation of initial DNA sample and generation of new 5'-PO4 ends by DNA shearing.
- For dephosphorylation we tested two phosphatase enzymes, one from Sigma and one from NEB. Both performed similarly, but complete dephosphorylation is infeasible. If we assume 1.5 ug of DNA, an average fragment length of 5000 bps, and a dephosphorylation efficiency of 99.9% we are still left with 288 million fragments of DNA that are not dephosphorylated. Given the low abundance of our target, we did not find

the significant amount of background DNA from background DNA from incomplete dephosphorylation reactions unreasonable. We used coverage with respect to untargeted sequencing as our comparison rather than % of reads of each run aligning to the target.

- At any given step in the workup after dephosphorylation, DNA shearing from sample handling will produce new 5'-PO4 ends. We aimed to preserve long fragments of DNA which are also the most susceptible to shearing. Various steps were taken to reduce DNA shearing during sample prep including: avoiding vortexing, avoiding excess pipetting, and using wide bore pipette tips when possible.
- The read lengths distributions observed are mainly the result of DNA extraction method. Multiple extraction methods were tested on poultry feces using read length and yield as our primary evaluation metrics. Poultry feces was selected as the sample type to optimize around because it can have a high uric acid concentration, interfering with DNA extraction efficiency. We tested the Cador Pathogen Kit and DNA PowerSoil Pro kits with different lysis procedures varying lysis vortexing duration and temperature. Bead beating was necessary for sufficient lysis in these sample matrices. The protocol described in the methods had the highest DNA yield using a qubit fluorimeter and DNA fragment lengths using a fragment analyzer. Average fragment lengths from the different methods resulted in 1000-5000 bps. Performance using the actual samples with the PowerSoil Pro Kit was variable by sample but similar to the expected average fragment length of 5000 bps from preliminary testing.

Revised text as it appears in text lines 373-377: “While the on-target read percentage was relatively low, the overall coverage of target genes was still high. The background DNA is likely the result of incomplete enzymatic reactions during library preparation. For example, a high dephosphorylation efficiency, even as high as 99.9%, would still result in millions of phosphorylated DNA fragments given our high input DNA concentration.”

3. In general, the methods read as a very detailed step-by-step protocol, but it seems to be missing the “why” for some steps. Consider reworking the methods (where possible) to be more generalized with a detailed, step-by-step protocol that can be published separately on [protocols.io](https://www.protocols.io). For example, if there is no deviation from the ONT ligation sequencing kit used from lines 411-420, then you can simplify the explanation. Many decisions were obviously made to troubleshoot and optimize this assay, and I think the more conversational information that is included with justifications, the better. Could have missed it, but a reasoning/purpose for the addition of Proteinase K is needed.

- We thank the reviewer for this comment. We added in more details to explain why certain steps were needed, including why Proteinase K was added. The volumes used in our protocol are based on Timp et al. which is already on [protocols.io](https://www.protocols.io) (<https://www.protocols.io/view/cas9-enrichment-for-nanopore-sequencing-5qpvondezl4o/v3>). Our protocol as well as the Timp et al. protocol deviates for some volumes from the ONT ligation sequencing kit and the ONT Cas9 sequencing kit thus we prefer to keep all the volumes as written. We included the [protocols.io](https://www.protocols.io) as a citation and have edited the methods section to include more details on why each step is needed.

Revised text as it appears in text lines 485-486: “Cas9 cut time is a balance of increased time for on-target binding and cutting but may increase off-target binding.”

Lines 488-489: “Multiple Cas9 guides are used for one region of interest in similar approaches.”

Lines 490-491: "Proteinase K was added to remove Cas9 with the hypothesis that Cas9 could block pores leading to early poor death."

Lines 500-501: "Dephosphorylation of the sample DNA is needed so only the phosphorylated ends of the DNA that result from Cas9 cutting are ligated to sequencing adapters."

Lines 504-505: "Cas9 cutting introduces blunt end cuts and A-tailing is used to create TA overhangs for adapter ligation."

4. For the adaptive sequencing, it reads as if only CTX-M-55 is used as a guide to capture both CTX-M and TEM variants. Please clarify.

- We thank the reviewer for this comment. We primarily used CTX-M as the model ARG for testing protocol modifications, unless the modification was specific to the impact of including more than one target, then we also included TEM. We used one ARG target in order to reduce the variables being changed at one time. We have clarified this in the text.

Revised text as it appears in text lines 475-476: "We primarily evaluated modifications for only CTX-M unless the modification was specific to multi-target detection"

Minor Comments/Suggestions:

5. Line 76: "overlap in AMR elements" is too vague. Consider using the concept of "genetic synteny" to describe this work throughout.

- Updated here and throughout to genetic synteny or genetic similarity.

Revised text as it appears in text line 37: "Using this method, termed Context-Seq, we investigated genetically similar..."

Line 78: "Current methods to investigate genetically similar AMR elements..."

Lines 101-102: "...to investigate genetic synteny in antimicrobial resistance elements"

Lines 331-332: "We observed significant genetic synteny between poultry and canines..."

Lines 553: "...all-versus-all was used to identify regions of genetic synteny between..."

6. Line 77-78: *Captures a small fraction of the population of a given species.

Is there ever a comparison to the sensitivity between the TaqMan assay and Context-Seq?

- We thank the reviewer for this suggestion. We did not compare sensitivity between Context-Seq and the qPCR array card (TAC) as the main goal of Context-Seq is understanding genomic context, not quantifying gene copies. In order to do this, we would need to spike in known copy numbers of the CTX-M and TEM genes based on qPCR of the isolate but that would likely require sequencing a series of dilutions. We did compare performance of Context-Seq to culturing from the same samples followed by whole genomic sequencing and also to untargeted/no enrichment of the samples.

No changes made.

7. Line 135: Is it accurate to describe this a “mock community”? How can you be sure that those ESBLs weren’t already present? Also, it reads as if whole cells were spiked into raw DNA extracts which threw me off before I read the supplemental.

- We thank the reviewer for bringing this up. We agree that mock community is not the most accurate term. We modified to “test system” to more accurately convey the optimization system in the text, and have also clarified that we spiked in DNA from the control strain.

Revised text as it appears in text lines 139-140: “We investigated these modifications on a test system comprised of DNA extracted from an *Escherichia coli* isolate ...”

8. Line 205-206: Denote coverage as “160x”.

- We have updated the text.

Revised text as it appears in text lines 244-245: “coverage was 160x with Illumina sequencing”

9. Line 303-304: This is very interesting. Were the sampled households in linked communities/compounds to encourage the inferred sharing? It may be worth noting the relative proximity of the sample households to one another (if I’ve missed that in text).

- We thank the reviewer for this comment and for bringing up this excellent point. Households 1 and 2 are from Kibera and households 3 and 4 are from Dagoratti South. Within the same community, the households are within 0.1 miles of each other. Between communities the households are 3 miles apart. Interestingly, we do not observe more sharing within than between communities (ie 1 & 2 vs. 3 & 4). We have included a discussion of this in the text.

Revised text as it appears in text lines 280-284: “Households one (HH 1) and two (HH 2) are located in Kibera while households three (HH 3) and four (HH 4) are located in Dagoretti South. Within the same community, households were approximately 0.1 miles apart while household in different communities were approximately 3 miles apart. 13 out of 23 clusters were shared between Kibera and Dagoretti South, five in Kibera only, and five in Dagoretti South only.”

Lines 344-345: “The paired study also identified that within household sharing was more common than between household sharing, yet we found that over half of sharing events occurred in both communities”

10. Line 305: When the phrase “ARG sharing” is used here it implies HGT, although cross-cell HGT is not typical via chromosomal ARGs. Maybe rephrase to say corresponding genetic contexts were observed between chromosomes and plasmids across samples?

- We rephrased this sentence.

Revised text as it appears in text line 353: “We observed similar ARGs and surrounding genomic context in both plasmids and chromosomes.”

11. 315-324: Given the exorbitant DNA requirements for ONT (1.5-3 ug), many environmental matrices, notably surface, ground, and drinking water, may be limited by the technique (notwithstanding newer library prep kit requirements). Need to define PAM, CrRNA, and tracrRNA.

- We have added in a discussion of the DNA requirement in environmental matrices and have defined these terms.

Revised text as it appears in text lines 399-401: “The feasibility of environmental matrices also varies as the input DNA requirement (1.5-3.0 µg) would require extensive concentration for some sample types (e.g. surface or drinking water).”

Line 459: “10 PAM(protospacer adjacent motif)-...”

Line 492: “CrRNAs (CRISPR RNA) and tracrRNAs (trans-activating CRISPR RNA),...”

12. 423-424: How long were the flow cells run?

- Flow cells were run for 72 hours or were stopped in the event of no available pores.

Revised text as it appears in text lines 524-525: “Samples were run on MinION (FLO-MIN106) R9.4.1 flow cells using a MK1B sequencer for 72 hours or until there were no available pores.”

13. Bioinformatics: Give parameters/settings used for each software.

- The methods text has been revised with the commands and non-default parameters if used.

Reviewer #2 (Remarks on code availability):

The code provides sufficient information for reproducibility but I did not attempt to install and test.

Reviewer #3

Fuhrmeister et al. provides an example of how targeted Nanopore sequencing with Cas9-guided adapter ligation can be used to characterise selected antimicrobial resistance genes within their genomic context. The authors demonstrate that chromosomally and plasmid-encoded blaTEM and blaCTX-M genes are shared between adults, children, poultry, and dogs in animal-owning households in Nairobi, Kenya. The manuscript is clearly presented and written.

I have a number of issues for the authors to consider:

1. The idea to use Nanopore Cas9-targeted sequencing to characterise antimicrobial resistance genes within their genomic context is not entirely new. For example, Cottingham et al. used the same method to screen for antimicrobial resistance genes (as well as other targets) in three human faecal samples spiked with *Klebsiella pneumoniae* at varying abundance (DOI: 10.1128/msystems.01413-24). The authors should mention this study (as well as other similar studies if they exist) and compare the methodologies and results.

- We thank the reviewer for bringing this to our attention and pointing out this study. We note that the study was not published when we submitted our manuscript. We added reference to it as well as three others in the introduction. We also added discussion comparing our method to those used in the Cottingham et al. study. Our methods differ in that we focused on specific, clinically important ARGs and split our library into two for sense and anti-sense cutting. To our knowledge, our study is the first to use Cas9 targeted sequencing in unspiked fecal samples in a field study setting.

Revised text as it appears in text Lines 88-90 “Recently, Cas9-based enrichment has been applied to short¹⁴ and long read sequencing for clinical applications^{15,16} targeting a variety of

genes including *Klebsiella pneumoniae*-associated genes¹⁷, ARGs¹⁸, cancer-associated genes¹⁵, and integrons¹⁹.”

Lines 378-387: “For example, a previous study investigated the detection of *K. pneumoniae*-associated genes, including CTX-M-15, using fecal samples spiked with variable colony forming units of *K. pneumoniae*.¹⁷ The authors did not identify any CTX-M reads in the untargeted samples but did identify 1-8 reads in the targeted, whereas we identified up to 419 reads as CTX-M. The previous work used a guide pool of 31 pairs of guides targeting many *K. pneumoniae* genes, while we used 2 guide pairs (one for each gene).¹⁷ This reduces the guides targeting CTX-M whereas half the guides in our pool targeted CTX-M genes. They also multiplexed samples, which is more cost effective but reduced the available pores and sequencing effort per sample. This previous work did demonstrate high coverage across *K. pneumoniae* genes,¹⁷ and together with our study, demonstrates the utility of Cas9-targeted sequencing.”

2. The authors selected two targets (*bla*TEM and *bla*CTX-M) and screened 13 faecal human, canine, and avian samples collected from four households that were positive for *bla*CTX-M group 1 and *bla*TEM. Despite the small sample size, the authors were able to identify genetically distinct clusters containing *bla*TEM and one cluster containing *bla*CTX-M that were shared within and between households. While humans and animals in close contact are already known to share bacteria and antimicrobial resistance genes, the study clearly demonstrates the usefulness and advantages of Nanopore Cas9-targeted sequencing compared to other methods. The authors also screened a mock community comprising an *Escherichia coli* isolate with *bla*CTX-M-55 and *bla*TEM-1 genes spiked into a composite sample of extracted DNA from Kenyan soil, but these results are not clearly presented. As the authors mention, it would have been interesting and very relevant in a One Health perspective to investigate whether Nanopore Cas9-targeted sequencing can be used as a tool to screen for antimicrobial resistance elements in soil and wastewater samples. This could be done by spiking soil and wastewater samples with the *Escherichia coli* isolate containing the *bla*CTX-M-55 and *bla*TEM-1 genes or by screening pristine soil and wastewater samples that are known to be positive for *bla*TEM and *bla*CTX-M (or other clinically relevant targets). This information would significantly increase the novelty and impact of the manuscript.

- We fully agree with the reviewer that soil and wastewater are relevant from a One Health perspective. Here, we selected soil as the test system to develop methods in as we thought it may be the most challenging matrix to sequence. As the reviewer suggested above, our method development included spiking DNA extracted from household soil samples from the study communities to test modifications of the protocol. Thus, our method development experiments confirms that the method can work on soil. Given the method also worked on human and poultry feces, we expect the method to also work on wastewater.
- We have added the following sentence in the methods section to highlight the use of soil in our method experimentation:

Added text in line 475: “Soil was chosen as the test matrix to confirm the method works on a complex sample type.”

3. Please explain what you mean by “clusters” in l. 39-40 and throughout the manuscript.

- We thank the reviewer for bringing up this point of confusion. We have introduced language in the methods earlier in the manuscript when clusters are first defined.

Revised text as it appears in text Line 39: “genetically distinct clusters (DNA sequences with > 80%ID over \geq 3000 bp)”

Lines 187-189 “As described in more detail in the methods, reads >1500 bps containing ARGs were clustered at 85% and polished if multiple reads were assigned to the same cluster.”

4. It is unclear to me what “pairs” refers to in l. 125.
 - We thank the reviewer for bringing this to our attention. We have updated the text to more clearly state this refers to the sense and antisense strand cutting.

Revised text as it appears in text Lines 132-133: “When comparing guides for *bla*_{TEM}, all pairs (sense and antisense) resulted in enrichment”

5. The sentence in l. 144-146 is unclear and should be rephrased.
 - We rephrased the text.

Revised text as it appears in text Lines 158-160: “There was a minor decrease in *bla*_{TEM} when comparing coverage between one-target and two-target enrichment. The non-normalized coverage (mean \pm std) was 1265 ± 253 for two targets and 1517 ± 337 for one target.”

6. What does “(35)” in l. 153 refer to? Is it the standard deviation? In that case, it is very high.
 - Correct, this does refer to the standard deviation. We clarified this in the text and added more discussion regarding the method and variable performance by sample.

Revised text as it appears in text Lines 169-170: “This generated one consensus sequence per cluster and resulted in an average of 39 ± 35 clusters per sample (Table S7).”

Lines 372-373: “Context-Seq also resulted in a significant increase in coverage of our target genes compared to untargeted sequencing, though performance varied by sample”

7. Please explain your findings in l. 209-211. Is it because *bla*_{TEM} and *bla*_{CTX-M} were located in another bacterial species than *Escherichia coli*?
 - The two samples are HH1 poultry and dog feces. In both cases (Figure 3A), *E. coli* was most commonly identified but other species were also present. Possible explanations for why these genes weren’t picked up by culture are both that other organisms were harboring the genes but more importantly the *E. coli* containing these genes were not selected for by the media. We have added this to a new paragraph in the discussion section.

Revised text as it appears in text lines 370-372: “Our target ARGs were identified in two samples with Context-Seq but not by culture, likely because the selective conditions of culturing enabled other strains to outcompete the ARG-containing bacteria.”

8. Chromosomal and plasmid sequences were screened for taxonomic associations by Kraken2 and BLASTN. Please state whether all sequences were associated with a single

bacterial species and discuss the potential confounding effect of interspecies horizontal gene transfer.

- We thank the reviewer for bringing this up and agree it is important to mention plasmids in regard to taxonomic identification and clarify the methods used for the taxonomic identification presented in the manuscript. We supplied an excel sheet (Supplementary Data file 1) that provides the top 5 matches in BLASTN for each consensus sequence and clarified our criteria when comparing BLASTN to kraken2.

Revised text as it appears in text Lines 393-395: “In addition, taxonomic identification of ARG hosts is challenging due to plasmid sharing, especially in Enterobacteriaceae.⁵² We only included identifications that were consistent across methods but in many cases reflects a *possible* host.” And See Supplementary Excel File 1

REVIEWERS' COMMENTS

Reviewer #2 (Remarks to the Author):

The authors have made substantial improvements to the clarity and justification of the manuscript and all methods. After review, I could only find one typo:

Line 448 of Track Changes Doc: *leading to early pore death*

Response: We thank the reviewer for noticing this. We have edited the text to correct this typo.

Reviewer #2 (Remarks on code availability):

The code is appropriately available and sufficient instructions are provided for testing and reproducibility.

Reviewer #3 (Remarks to the Author):

Review of NCOMMS-24-62079A

The authors have addressed all my issues and I am satisfied with the revision.